# Differential Expression of Mitochondrial Biogenesis Markers in Mouse and Human SHH-Subtype Medulloblastoma

**DOI:** 10.3390/cells8030216

**Published:** 2019-03-05

**Authors:** Maria Łastowska, Agnieszka Karkucińska-Więckowska, James A. Waschek, Paweł Niewiadomski

**Affiliations:** 1Department of Pathology, The Children’s Memorial Health Institute, Dzieci Polskich 20, 04-730 Warsaw, Poland; m.lastowska@ipczd.pl (M.Ł.); a.karkucinska-wieckowska@ipczd.pl (A.K.-W.); 2Department of Experimental and Clinical Neuropathology, Mossakowski Medical Research Centre Polish Academy of Sciences, A. Pawińskiego 5, 02-106 Warsaw, Poland; 3Intellectual Development and Disabilities Research Center, The David Geffen School of Medicine, University of California, Los Angeles, CA 90095, USA; 4Centre of New Technologies, University of Warsaw, Banacha 2c, 02-097 Warsaw, Poland

**Keywords:** medulloblastoma, Sonic hedgehog, Shh, oxidative phosphorylation, mitochondria, glycolysis, Warburg effect, gene expression, mouse model

## Abstract

Medulloblastoma is a brain tumor that arises predominantly in infants and children. It is the most common pediatric brain malignancy. Around 25% of medulloblastomas are driven by constitutive activation of the Hedgehog signaling pathway. Hedgehog-driven medulloblastoma is often studied in the laboratory using genetic mouse models with overactive Hedgehog signaling, which recapitulate many of the pathological features of human Hedgehog-dependent tumors. However, we show here that on a molecular level the human and mouse HH-dependent MB are quite distinct, with human, but not mouse, tumors characterized by the presence of markers of increased oxidative phosphorylation and mitochondrial biogenesis. The latter suggests that, unlike for many other types of tumors, a switch to glycolytic metabolism might not be co-opted by human SHH-MB to perpetuate their survival and growth. This needs to be taken into consideration and could potentially be exploited in the design of therapies.

## 1. Introduction

Medulloblastoma (MB) is the most common pediatric brain malignancy [1]. Up to 80% of cases are tractable with a combination of surgery and radio- and chemotherapy, but these invasive, non-targeted approaches often leave young patients with severe neurodevelopmental deficits [2]. For these reasons, a quest for more targeted therapies with fewer side effects is currently underway. This endeavor was made possible thanks to the recent studies showing that MB is molecularly diverse and comprises four distinct classes of tumors that differ in their mutation and gene expression landscape, pathological presentation, and prognosis [1]. These classes are: SHH-MB characterized by activation of the Hedgehog (HH) signaling pathway, WNT-MB characterized by high activity of WNT signaling, and two molecularly distinct groups named Group 3 and Group 4, where neither HH nor WNT signaling is activated. The SHH-MB group, comprising approximately 25% of all MB tumors, is the best characterized of the MB classes due to the ready availability of mouse models of the disease [3].

In recent years, inhibitors of HH signaling have emerged as some of the most promising anti-cancer therapeutics [4,5]. These inhibitors predominantly bind to and block the activity of SMO, a membrane protein that is the key signal transducer in the HH pathway. It was hoped that SMO inhibitors would constitute a new weapon against MB tumors in humans. However, it turned out that MB cells can develop resistance to SMO inhibitors [6]. Moreover, many cases of SHH-MB are inherently independent of SMO because they are triggered by mutations downstream of SMO in the HH pathway [7]. Consequently, SMO inhibitor drugs (Vismodegib, Erismodegib) were approved by the FDA only for the treatment of another HH-dependent cancer, basal cell carcinoma, but not SHH-MB.

Currently, the search continues for new drugs that would be effective in treating SHH-MB. Unfortunately, only two human SHH-MB cell lines are available commercially (DAOY, ONS-76) [8]. Thus, one of the instrumental tools for understanding the pathogenesis of human SHH-MB and developing anti-SHH drugs are genetically-modified mouse models of SHH-MB. A number of mouse models of SHH-MB were developed in the last two decades [3]. Included are those based on loss of the tumor suppressor Patched [9], gain-of-function of the oncogene Smoothened [10,11], or the conditional loss of the Gαs subunit of the heterotrimeric G proteins [12]. Despite their widely regarded utility as tools for cancer drug discovery and testing, recent work showed that mouse models do not faithfully reflect the molecular composition and pathological features of human SHH-MB [13].

Here, we analyze the gene expression profiles of multiple SHH-MB mouse models and compare them to those of human SHH-MB samples. We find that human SHH-MBs, unlike tumors in mouse models of the disease, are characterized by relatively high expression of genes associated with mitochondrial oxidative phosphorylation. Our work suggests that therapies that target the balance between aerobic respiration and glycolysis in tumor cells may have different outcomes in human and mouse SHH-MB.

## 2. Materials and Methods

### 2.1. Cerebellar Granule Cell Progenitor Culture

Cerebellar granule cell progenitors (cGCPs) were isolated and cultured as described [14]. Briefly, P3 mice were sacrificed by pentobarbital injection and decapitation. Cerebella were isolated and digested in a trypsin/DNAse solution. cGCPs were separated by centrifugation on a 35%/60% Percoll step gradient. They were cultured in media containing Neurobasal, penicillin/streptomycin/glutamate (1×), 1 mM sodium pyruvate, 0.04 μg/mL triiodothyronine, 0.1 mg/mL BSA, 0.04 μg/mL sodium selenite, 60 μg/mL N-acetylcysteine, 0.06 μg/mL progesterone, 5 μg/mL insulin, 100 μg/mL apo-transferrin, and 16 μg/mL putrescine. For real-time RT-qPCR analysis, the cells were treated for 6 h with or without Shh-N (1 μg/mL; R&D, Minneapolis, MN, USA), PACAP-38 (10 nM; Calbiochem, San Diego, CA, USA), or both.

### 2.2. Real-Time RT-qPCR

RNA was extracted using TRIzol. For cell culture, media was removed, and the TRIzol reagent was added directly to the wells. Tissues were homogenized in TRIzol using a rotor/stator homogenizer. RNA extraction was afterwards performed according to the TRIzol manufacturer’s protocol. cDNA was generated using the iScript cDNA synthesis kit (Bio-Rad, Hercules, CA, USA) and quantitative PCR was performed using the iQ SYBR green supermix (Bio-Rad). The results were analyzed using the standard curve method with GAPDH used as the housekeeping gene. Primers used for qPCR were as follows:


ForwardReverseGAPDHggccttccgtgttcctactgtcatcatacttggcaggttGli1atctctctttcctcctcctcccgaggctggcatcagaaGpr153ctcagagcctgccagaacttaagctcaccaccagcacagShisa2ctcggcagtccccatctaccgtagacatcggcaacagc

### 2.3. Animal Husbandry

Mice were housed according to UCLA Institutional Guidelines, including standard light–dark cycles with food and water ad libitum. All mice were daily monitored for signs of illness and sacrificed when visibly unwell. WT, *Ptch1* [9] and *PACAP*^+/−^/*Ptch1*^+/−^ [15] mice were generated and maintained as described previously.

### 2.4. In-Situ Hybridization

P6 mouse brains were surgically removed and placed into 4% paraformaldehyde for 24 h for fixation. Adult tumor-bearing mice were transcardially perfused with 4% paraformaldehyde and brains were postfixed in 4% paraformaldehyde for 2 h. Tissues were cryoprotected in 20% sucrose and embedded in OCT media. Tissue was sectioned saggitally 10–12 μm and stored at −80 °C. In situ hybridization with digoxygenin-labeled probes was performed as previously described [16].

Templates for ISH probe synthesis were amplified from neonatal mouse brain cDNA using PCR and cloned into pCRII-TOPO plasmids. The probes were designed to bind region 2422-2758 of Gpr153 (NM178406.2) and region 1845-2195 of Shisa2 (NM145462.4). Both sense (S) and antisense (AS) probes were prepared by in vitro transcription using DIG nucleic acid detection kit (Roche, Basel, Switzerland) and tested following the manufacturer’s protocol.

Tissue sections were rehydrated in PBS with 0.1% Tween 20 and postfixed in 4% paraformaldehyde for 10 min. Then the sections were treated with proteinase K at 37 °C for 3 min, postfixed again in 4% paraformaldehyde for 20 min and hybridized with digoxigenin-labeled probes against Gpr153 (600 ng/mL) and Shisa2 (300 ng/mL) overnight at 55 °C. After hybridization, the sections were washed in a solution consisting of 50% formamide and 5× SSC and 1% sodium dodecyl sulfate once for 30 min at 60 °C, then twice in 50% formamide and 2× SSC for 45 min at 60 °C and finally in Tris-buffered saline containing 0.1% Tween 20 (TBST). The slides were then immersed in blocking solution (Roche) and incubated overnight at 4 °C with anti-digoxigenin Fab-alkaline phosphatase conjugate (Roche) diluted to 1:2000 by blocking solution. The hybrids were visualized by an alkaline phosphatase reaction using the NBT-BCIP method (Wako, San Diego, CA, USA). The slides were then dehydrated and coverslipped using permount.

### 2.5. Analysis of Medulloblastoma Gene Expression Datasets

All data analysis was done in R. For each of the datasets, the “matrix” file was downloaded from GEO and imported into R using the getGEO function from the GEOquery library [17]. If necessary, data were renormalized using the invariant set method (normalize.ExpressionSet.invariantset from the affyPLM library [18]). Plots of individual expression log2 values were drawn using the beeswarm library using the data from the probe with the maximum mean log2 expression for each gene if more than one probe matched that gene.

For calculations involving differentiation between *TP53* wild-type (WT) and *TP53* mutant SHH-MB tumors, the information regarding the mutation status of the *TP53* gene was extracted from Appendix A from [7] and cross-referenced with tumor identifiers in the dataset GSE49243. Only data from tumors where *TP53* was sequenced was included in graphs and statistical significance calculations.

To select genes that showed the highest difference in expression between human and mouse tumors, we applied the following procedure. First, for each probeset in each microarray dataset, we calculated median expression value for this probeset in each of the tumor/tissue subtypes. This generated a table with probesets in rows and tumor/tissue types in columns. In the next step, we used the collapseRows (“MaxMean” method) from the WGCNA library [19] to select the most highly representative probeset for each gene, which resulted in a table with genes in rows and tumor/tissue types in columns. Next, we normalized each row by subtracting the mean value for that row from all values within the row (normalized median gene expression values). For human datasets, the columns typically represented different subtypes of MB, whereas for mouse datasets, the columns included normal cerebellum as controls. This generated data that allowed us to determine whether the median expression of a gene in a specific tumor/tissue type is higher (positive values) or lower (negative values) from other tumor/tissue types in the same dataset (tumor/tissue-dependent overexpression values). We then ordered genes for each dataset according to their overexpression values in the SHH-MB/Shh-MB group and calculated quantile ranks. These ranks were averaged separately for mouse Shh-MB and human SHH-MB groups. Genes with high ranks (closer to 1) in human tumors, but low ranks (closer to 0) in mouse tumors were considered to be human SHH-MB-specific, and genes with low ranks in human tumors and high ranks in mouse tumors were considered to be mouse Shh-MB-specific.

Of note, datasets containing gene expression for human samples do not contain healthy cerebellum controls, whereas all mouse datasets do contain healthy samples as controls. To ensure that the choice of controls does not affect analysis results, we repeated gene ranking using a recently published combined dataset of gene expression results from healthy cerebella and different medulloblastoma subtypes available from the GEO accession number GSE124814 [20]. The analysis was performed as follows. For each gene and each medulloblastoma subgroup or cerebellar control, a median log-transformed expression value was calculated. The cerebellum control medians were then subtracted from median log expression values for each medulloblastoma subgroup, which yielded cerebellum-normalized median log expression values, which were used for gene ranking. Similarly, for each mouse dataset, a median log-transformed expression was calculated for each gene and each medulloblastoma subgroup or cerebellar controls, and the cerebellum control median was subtracted from all other groups. Cerebellum-normalized median log expression values for Shh-MB were then averaged across mouse datasets and used for subsequent gene ranking.

Source code and raw/processed data is available upon request.

### 2.6. Gene Set Enrichment Analysis

To discover functional groups of genes that were either mouse Shh-MB specific or human SHH-MB specific, genes were ordered according to the difference between ranks in human and mouse SHH-MB tumors and the GSEApreranked tool was used [21]. The following groups of gene sets from the MSigDB database [21] were used in the analysis: h.all.v6.2.symbols.gmt (hallmark gene sets), c2.all.v6.2.symbols.gmt (curated gene sets), c5.all.v6.2.symbols.gmt (GO gene sets).

### 2.7. Immunohistochemistry

The analysis was performed on formalin-fixed paraffin embedded (FFPE) tissue samples. Expression of COX4 protein (cytochrome c oxidase subunit 4) was detected using antibody clone F-8 (Santa Cruz Biotechnology Inc., Santa Cruz, CA, USA code: sc-376731, dilution 1:200). Antigen retrieval was performed using Target Retrieval Solution, Low pH, (DAKO, Glostrup, Denmark) for 30 min in 99.5 °C. Whole preparations were scanned in Hamamatsu NanoZoomer 2.0 RS scanner (Hamamatsu Photonics, Hamamatsu, Japan) at the original magnification 40×. p53 IHC was performed on the Ventana BenchMark ULTRA IHC/ISH autostaining system using a mouse monoclonal antibody (BP53-11) after antigen retrieval in CC1 buffer followed by detection with the Ultra View HRP system (Roche/Ventana, Basel, Switzerland).

Molecular classification of medulloblastoma samples was determined and described previously [22]. Briefly, NanoString nCounter system analysis (NanoString Technologies, Seattle, WA, USA) was applied for identification of 4 molecular groups (WNT, SHH, Group 3, or Group 4) using a NanoString CodeSet of 22 marker genes and 3 housekeeping genes (ACTB, GAPDH, and LDHA). Raw counts for each gene underwent technical and biological normalization using nSolver 2.5 software (NanoString Technologies, Seattle, WA, USA). The clustering of samples was performed with Pearson correlation and average settings.

Four SHH tumors were determined by the presence of an immunohistochemical positive reaction with anti-GAB1 (Abcam, #ab27439 and/or ab #59362, dilution 1:100) and anti-YAP1 (Santa Cruz Biotechnology, #sc-101199, dilution 1:50) antibodies, as previously described [23].

## 3. Results

### 3.1. Gpr153 and Shisa2 are Shh Targets in Murine Cerebellar Granule Cell Progenitors and Mouse Shh-Type Medulloblastoma

SHH-MB is thought to arise from cerebellar granule cell progenitors (cGCP) [24], which proliferate extensively in the early postnatal period in a manner dependent on the Hh pathway. The proliferation of cGCPs and SHH-MB formation is antagonized by receptors coupled to the small GTPase Gαs, which stimulates the activity of protein kinase A and directly inhibits the Hh-dependent transcription factors belonging to the Gli family. Consequently, the deletion of Gαs in cGCPs results in SHH-MB formation [12]. Moreover, we previously showed that both the Shh-dependent proliferation of cGCPs and Shh-MB tumorigenesis can be blocked by the neuropeptide PACAP (*Adcyap1*), one of the cerebellar ligands of Gαs-coupled receptors. Tumors arise more quickly and with a much higher incidence in *Ptch1*^+/−^ mice that also lack a single *Adcyap1* allele [14,15,25]. To identify novel biomarkers and potential therapy targets for human SHH-MB, we previously profiled gene expression genome-wide in cGCPs treated with Shh or co-treated with Shh and PACAP [14]. We hypothesized that genes that responded to both physiological stimuli, i.e., were upregulated in Shh-treated cGCPs but were downregulated by the addition of PACAP could serve as markers of Shh/Gli activity in MB precursor cells and potential biomarkers/therapy targets for human SHH-MB. Two genes in particular were very strongly upregulated in Shh-treated, but not Shh+PACAP-treated, cGCPs: *Gpr153* and *Shisa2* (Figure 1A; the full set of regulated genes is available as a supplement for reference [14]). These genes were of particular interest to us because they encode transmembrane proteins that might be suitable as easily accessible potential therapy targets or biomarkers. One of these genes, *Gpr153*, was previously reported to be upregulated in Shh-treated cGCPs and in mouse Shh-MB and to be a direct target of Gli1 (assessed by chromatin immunoprecipitation) [16]. *Shisa2* had not been reported as a HH target gene before and is likely not a direct Gli protein target, since blocking new protein synthesis using cycloheximide attenuates the effects of Shh on *Shisa2* expression (Appendix A).

We found that the known HH target genes *Gli1* and *Gpr153*, as well as our newly discovered HH target gene *Shisa2*, were upregulated in Shh-MB in the mouse. First, we compared the expression of putative Shh target genes in wild-type young (postnatal day 4; P4) and adult cerebella as well as two types of Shh-MB tumors—those derived from *Ptch1*^+/−^ mice and those derived from mice heterozygous for both *Ptch1* and *Adcyap1* (double heterozygous; DHz) [15]. We found that all three putative HH target genes were expressed at higher levels in young highly proliferative cerebella compared to adult mature cerebella. All three were also strongly upregulated in murine Shh-MB tumors from *Ptch1*^+/−^ and DHz mice (Figure 1B). These results were further confirmed using in situ hybridization (Figure 1C,D).

### 3.2. Gpr153 and Shisa2 Are Not Upregulated in Human SHH-MB

Based on encouraging results from the mouse model, we expected that Gpr153 and Shisa2 might serve as biomarkers in human SHH-MB. Because they are both transmembrane proteins, they might also be used as conveniently accessible drug targets in the clinic. We analyzed the expression of *GPR153* and *SHISA2* across a set of 285 human MB tumors, including 51 SHH-MB samples (microarray analysis results available from GEO—dataset GSE37382) [26]. Surprisingly, neither *GPR153* nor *SHISA2* expression was elevated in human SHH-MB samples compared to Group 3 and Group 4 tumors (WNT type tumors were not profiled in this dataset). In contrast, *GLI1* was highly overexpressed in human SHH-MB, as expected. The lack of *GPR153* and *SHISA2* overexpression in human SHH-MB was further confirmed using another human MB gene expression dataset available from GEO [27] (GSE10327; Figure 2A).

Given this discrepancy in gene expression between human SHH-MB and mouse Shh-MB, we wanted to know if *Gpr153* and *Shisa2* overexpression was a general feature of Shh subtype MBs in mice or whether they were limited to a single model with heterozygous deletion of Ptch1. We analyzed available datasets from multiple laboratories derived from four different models of Shh-MB: *Ptch1*^+/−^, CAGGS-CreER; R26-SmoM2, hGFAP:GnasCKO, Oligo1:GnasCKO [12,28,29,30] (Table 1). Expression of *Gpr153* and *Shisa2* was elevated in Shh-MB tumors independent of the genetic model used and independent of controls (normal early postnatal cerebellar tissue/prenatal cerebellar anlage/MB tumors from a different subgroup; Figure 2B). Thus, increased expression of *Gpr153* and *Shisa2* is a characteristic feature of mouse Shh-MB tumors, but not human SHH-MBs.

### 3.3. Mouse Shh-MB Have Different Gene Expression Profiles from Human SHH-MB

It has been shown that human and mouse Shh-MB tumors are distinct in terms of their gene expression profiles [13], but the conclusions of this study were based on a single microarray platform used for each species (HGU133 plus 2.0 for human, Mouse Genome 430 2.0 form mouse). To determine whether differences in gene expression between mouse and human Shh-MB are consistent independently of microarray methodology, we analyzed multiple datasets from GEO that had been generated on various microarray platforms for mouse and human (Table 1) [12,26,27,28,30,33]. We calculated differences between median log-transformed gene expression values for Shh-MB and the average median expression for all groups in each dataset. We then quantile-ranked genes that were represented in all microarray platforms based on these differences for each dataset (Appendix A). As expected, known universal Shh target genes ranked near the top among genes upregulated in Shh-MB for all datasets, whereas *Gpr153/GPR153* only ranked near the top in mouse datasets (Figure 3A). Overall, the ranks of Shh-MB gene expression changes were highly correlated among different human datasets and among different mouse datasets but were less well correlated between human and mouse datasets (Figure 3B) independently of microarray platforms and specific mouse Shh-MB models.

### 3.4. Expression of HH Target Genes Is Similar Between Mouse and Human SHH-MB

The differences in gene expression profiles between SHH-MB in human disease and in mouse models can be due to two different factors. First, the wiring of the HH pathway-dependent transcriptome might be significantly different between human and mouse. Second, the differences may stem primarily from differences in pathophysiology between mouse and human tumors independent of the driving mutations in the HH pathway. If the differences result from different HH-pathway wiring, we would expect mouse Shh-MB-upregulated genes to overlap with direct and indirect HH target genes in the mouse, whereas human SHH-MB upregulated genes would show no such correlation with mouse HH targets. To test this hypothesis, we compared the ranks of genes in the mouse Shh-MB and human SHH-MB datasets with genes ranked by their coexpression with mouse *Gli1* obtained from the coxpresdb dataset [35,36]. As expected, the ranks of mouse Shh-MB-upregulated genes showed significant correlation with the ranks of *Gli1*-coexpressed genes (Figure 4A). Interestingly, genes upregulated in human SHH-MB were also enriched for genes coexpressed with mouse *Gli1* (Figure 4B), suggesting that the wiring of the HH pathway transcriptome itself is similar in human and mouse tumors. We thus conclude that the differences in gene expression profile between human and mouse SHH-MB tumors reflect differences in tumor pathophysiology that are at least partially independent of the HH pathway itself.

### 3.5. Human SHH-MB, But Not Mouse Shh-MB, Have High Expression of Genes Associated with Mitochondrial Oxidative Phosphorylation

To determine what specific features of tumor pathophysiology determines the differences between mouse and human SHH-MB tumors, we ranked genes by differences between their overexpression values between human and mouse tumors, with highest positive values for genes that were highly upregulated in human SHH-MB tumors but downregulated in mouse Shh-MB tumors, and lowest negative values for genes that showed upregulation in mouse but downregulation in human SHH-MB. To determine what physiological processes were associated with the human but not mouse SHH-MB and vice versa, this list of genes was then fed into the Gene Set Enrichment Analysis (GSEA preranked) tool [21]. The results of GSEA point to genes associated with mitochondrial biogenesis and oxidative phosphorylation as those that, as a group, show the highest upregulation in human SHH-MB but are not upregulated in mouse Shh-MB (Table 2, Figure 5A,C). On the other hand, genes upregulated in mouse Shh-MB but not human SHH-MB are associated with active cell cycle and the Rb-E2F pathway (Table 3, Figure 5B,D).

Importantly, the human MB datasets available typically do not include healthy cerebellum controls. For this reason, we used MB samples belonging to other subgroups as controls in our analysis of human datasets. To ensure that our results are not biased by this choice of control samples, we repeated gene ranking using a recently published dataset in which a novel statistical method was used to combine human SHH-MB expression datasets with datasets containing expression data for healthy human cerebella [20]. Importantly, our calculated differences in gene ranks between human and mouse SHH-MB are mostly independent of the choice of controls (Appendix A). Moreover, when GSEA was performed on cerebellum-normalized gene ranks, the gene sets that were enriched in human and mouse SHH-MB samples were to a large extent similar to those obtained in our original analysis (Appendix A). Crucially, oxidative phosphorylation-related genes were on average upregulated in human SHH-MB compared to human cerebella and were strongly downregulated in mouse Shh-MB compared to mouse cerebella (Figure 5C,D and Appendix A).

We then picked a few of the genes in the oxidative phosphorylation/respiratory electron transport category for a more detailed expression analysis in selected datasets. (Figure 6). The expression of these genes is increased in human SHH-MB tumors compared to the other subgroups and is either decreased or unchanged in mouse Shh-MB compared to normal adult or neonatal cerebellum. Because adult/childhood SHH-MB was found to be more similar to mouse Shh-MB models than SHH-MB in infants [13], we hypothesized that perhaps the increased expression of mitochondrial chain genes in human SHH-MB would only occur in infant SHH-MB, but not in the adult/childhood cases. However, we found no consistent significant differences in expression of these genes among different age groups of SHH-MB samples (Figure 7).

### 3.6. Master Regulators of Mitochondrial Biogenesis Are Overexpressed in Human SHH-MB

One of the ways cells regulate the balance of oxidative phosphorylation vs glycolysis is via so-called “master regulators” of mitochondrial biogenesis: PGC-1α and PGC-1β [37]. In melanoma, PGC-1α expression is in turn controlled via the transcription factor MITF [38]. To determine whether the MITF->PGC-1 axis may be involved in the increased expression of respiratory chain genes in human SHH-MB, we checked the expression of genes encoding these proteins (*MITF*, *PPARGC1A*, and *PPARGC1B*) in medulloblastoma transcriptomic datasets. Interestingly, all three factors are upregulated in human SHH-MB, but their murine homologs are not upregulated in mouse Shh-MB (Figure 8), suggesting that the increase in genes encoding respiratory chain proteins in human SHH-MB is due to the upregulation in master regulators of oxidative phosphorylation from the PGC-1 family, and their upstream regulator MITF.

### 3.7. COX4 Is Highly Expressed in Human p53-Negative SHH-MB

To check if transcriptomic gene expression measurements are reflected in protein expression levels, we performed immunohistochemistry on tissue sections from human MB tumors of different molecular subtypes resected at the Children’s Memorial Health Institute in Warsaw, Poland. In agreement with transcriptomic data, SHH-MB tended to be strongly positive for COX4, one of the mitochondrial respiratory chain markers, but only if the tumor was not immunoreactive for p53 (Figure 9A). Most p53-positive tumors in the SHH subgroup had low immunoreactivity for COX4. COX4 immunoreactivity in WNT subgroup tumors, as well as Group 3 and Group 4 tumors, was overall lower than in p53-negative SHH-MB (Figure 9C,D). Since strong immunoreactivity for p53 is often used as a surrogate marker for *TP53* mutations [39], our data suggest that high expression of respiratory chain components is negatively correlated with mutations in *TP53* in SHH-MB tumors. Interestingly, we also found very strong COX4 immunoreactivity in one sample of MBEN (medulloblastoma with extensive nodularity)-type SHH-MB in the internodular region of the tumor (Figure 9B).

To check if *TP53* mutations may be associated with lower expression of oxidative phosphorylation genes in SHH-MB, we analyzed a gene expression dataset for SHH-MB for which matched genomic DNA analysis is also available [7]. Out of 73 microarray-analyzed tumor samples, we could find matching information on *TP53* mutations on 30 of these. Twenty-five were *TP53* wild-type, and 5 had *TP53* mutations, including 3 germline mutations, 1 somatic mutation, and 1 mutation where the origin was unknown. Although the expression of *COX4I1*, *COX6A1*, and *NDUFB5* tended to be lower in *TP53*-mutated tumors (Figure 10A), the differences were not statistically significant. Glycolysis markers (*GAPDH*, *HK2*, *PFKM*), on the other hand, tended to be increased in *TP53* mutant tumors with two of them (*HK2* and *PFKM*) having statistically significantly higher expression in the *TP53* mutant group (Figure 10B). We are hesitant to make strong conclusions from this data due to the limited sample size, but we believe that further analysis on a larger set of samples with matched genomic/transcriptomic data is warranted.

## 4. Discussion

Animal models of disease constitute an indispensable step in testing novel targeted therapies against cancer [40]. Specifically, genetically modified mouse models of MB have been a staple among translational researchers developing new drugs targeting this disease. Models of SHH-MB based on the deletion of *Ptch1* or overexpression of constitutively active Smo are relatively straightforward to maintain and give rise to tumors in young animals [9,10,11]. Despite the routine use of transgenic mice in testing new drugs against SHH-MB, the translation from the mouse model to treatment of human disease has been relatively disappointing with few notable exceptions. The major reason for the clinical failure of treatments developed in mouse models are profound differences between the heterogeneous disease in humans and its “clean” model in the murine host [13].

When studying both in vitro and in vivo models of SHH-MB, we discovered that the gene expression profile in mouse MB tumors and their human counterparts differs significantly. One major difference that we discovered was that human SHH-MB tumors have high expression of genes associated with the mitochondrial respiratory chain and oxidative phosphorylation, while mouse Shh-MB models do not. This was consistent across several patient cohorts and disparate studies and across several murine MB models.

In addition to having reduced expression of mitochondrial genes, mouse SHH-MB also appears to be more highly proliferative, as suggested by elevated expression of cell cycle-associated genes. It is tempting to speculate that the higher proliferation rate may be associated with a drop in oxidative phosphorylation in mouse Shh-MB. Consistent with this hypothesis, immunohistochemical staining of human MB samples showed that mitochondria are abundant mostly in SHH-MB negative for p53, a surrogate marker of *TP53* mutations [41,42,43]. This suggests that the more aggressive *TP53* mutant SHH-MB [44,45,46,47] shift towards more glycolytic metabolism, like the highly proliferative mouse Shh-MB tumors. Our attempt to corroborate this data with matched genomic/transcriptomic information on SHH-MB samples did not produce conclusive differences. While strong conclusions are not possible at this point due to limited sample size, large variability among samples, and the inherent limitations of semi-quantitative immunostaining methods, these results warrant further study of the relationship between cell proliferation, oxidative phosphorylation, and *TP53* status in SHH-MB. Importantly, the negative correlation between *TP53* mutations and oxidative phosphorylation has been shown in other types of cancer [48,49,50], further strengthening the case for a more exhaustive investigation. Large-scale genomic, epigenomic, and transcriptomic studies of MB are already underway and have helped in developing more fine-grained stratification of patient cohorts [45,46]. These and similar studies will make it possible to comprehensively test the relationship between genotype, molecular subgroup, and metabolic state of human MB.

Our data cast some doubt on the utility of genetically-modified mouse models of SHH-MB for preclinical studies. However, the few alternative preclinical models have their own serious drawbacks [8]. MB cell lines cultured in vitro undergo selective pressure and show evidence of phenotypic and genetic drift [51]. Short term patient-derived cell cultures are heterogeneous and, like immortalized cell lines, rapidly adapt to unnatural monolayer culture conditions, with non-physiological extracellular matrix, oxygen levels, and nutrient availability. Different types of 3D cultures, either as spheroids or embedded in ECM matrices are not in wide-spread use but are likely to be more physiologically-relevant [52]. Alternatively, patient-derived orthotopic xenografts could be used as more physiological preclinical models, but they, too, show evidence of drift away from the original phenotype of the parent tumor [53]. A systematic characterization of different preclinical MB models is urgently needed to aid in the design of targeted therapeutic interventions in MB [8].

An important implication of our study for the therapy of MB is that glycolytic metabolism is not as common in human SHH-MB as it is in mouse models of the disease. Even though human MB tumors were shown to be characterized by high uptake of 18FDG, a hallmark of aerobic glycolysis [54,55], these studies did not take molecular subgroup into account. In the mouse, aggressive Shh-MB shows high 18FDG uptake. Moreover, inhibition of aerobic glycolysis through genetic ablation of *Hk2* markedly reduces the malignancy of these tumors [56]. This dependence of tumor cells on the Warburg effect and aerobic glycolysis and/or lipogenesis has been suggested as a potential therapy target in SHH-MB [57,58]. Nevertheless, data on the dependence of human SHH-MB cells on aerobic glycolysis is lacking. Our results show that a strategy exploiting glycolytic metabolism may not be efficacious in many human patients, but may be a good option for the more aggressive and treatment-resistant *TP53* mutant human tumors [7,44], which, like the aggressive mouse tumors, are characterized by high expression of *HK2* (Figure 10). Importantly, aerobic glycolysis and oxidative phosphorylation may coexist in some tumors, and the high expression of mitochondrial genes does not necessarily preclude the use of glycolytic mechanisms of energy generation. Therefore, staining for p53, COX4, and HK2, combined with more direct assays of metabolic states such as measuring lactate production, may help stratify patients for therapeutic intervention, especially until more exhaustive genomic and transcriptomic diagnostics become routine in clinical practice.

## Figures and Tables

**Figure 1 cells-08-00216-f001:**
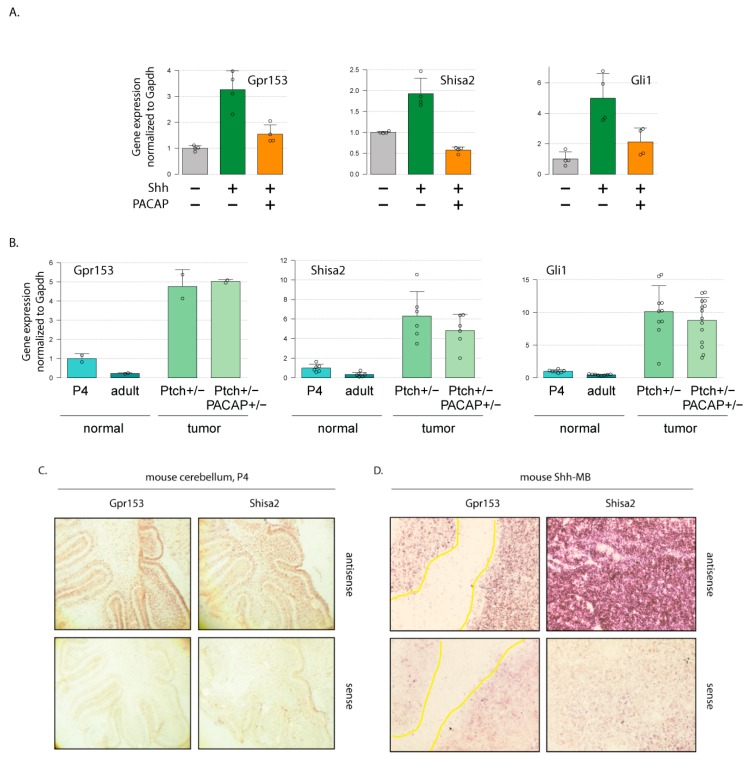
*Shisa2* and *Gpr153* are expressed in Shh-stimulated cerebellar granule cell progenitors (cGCPs) and murine Shh-MB. (**A**) Cultured cGCPs were treated for 6 h with 1μg/mL Shh-N and/or 10 nM PACAP38. Shown is the expression of indicated genes by RT-qPCR (mean +/− SD) from n = 4 independent samples/group. (**B**) Expression of indicated genes in healthy P4 and adult cerebella from WT mice, as well as Shh-MB tumors collected from *Ptch1*^+/−^ and *Ptch*^+/−^; *Adcyap1*^+/−^ (DHz) mice; shown are mean +/− SD for n = 11 samples per group. (**C**,**D**) Expression of *Gpr153* and *Shisa2* was measured by in situ hybridization on samples from P4 wild-type cerebella (**C**) and tumors collected from *Ptch*^+/−^ mice (**D**). Representative images are shown for hybridization of antisense (top) and control sense (bottom) probes.

**Figure 2 cells-08-00216-f002:**
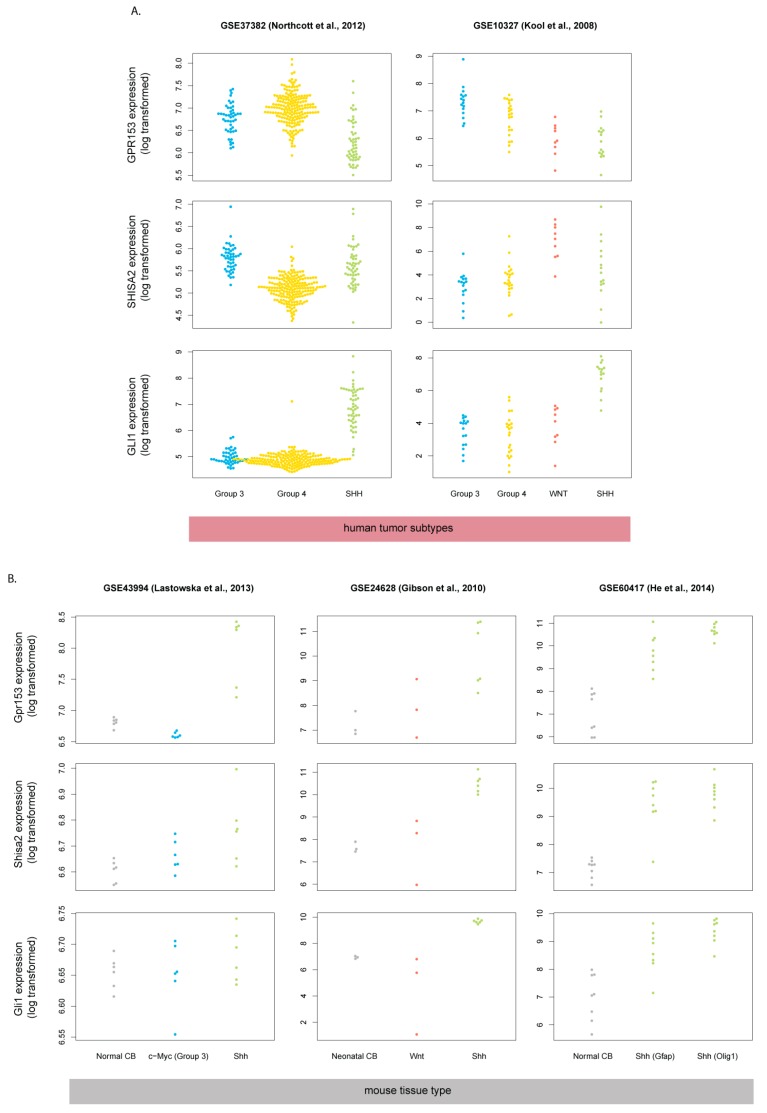
*GPR153* and *SHISA2* are not upregulated in human SHH-MB. Expression data derived from microarray analysis were downloaded from GEO (accession numbers and corresponding citations are shown) and reanalyzed. *Gli1/GLI1* is upregulated in both human (**A**) and mouse (**B**) Shh-MB samples, whereas overexpression of *Gpr153/GPR153* and *Shisa2/SHISA2* is limited to mouse Shh-MB.

**Figure 3 cells-08-00216-f003:**
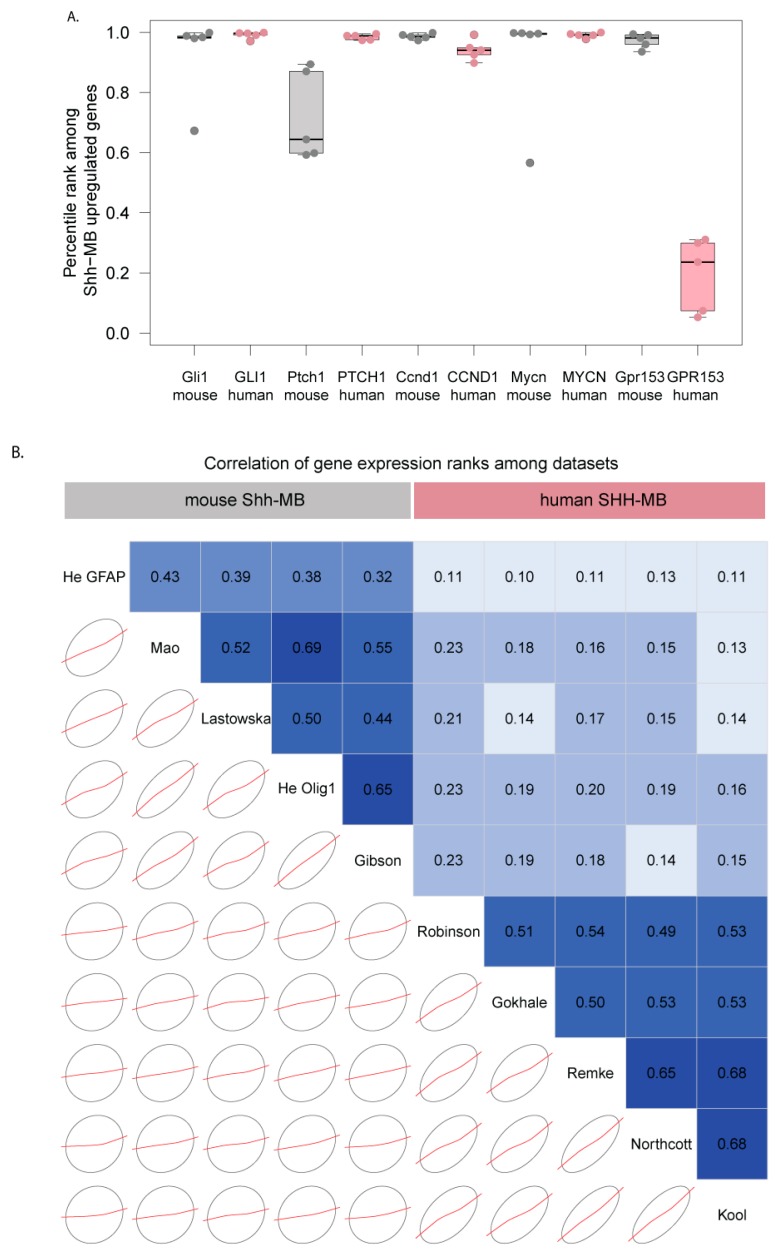
(**A**) *GPR153* differs in expression from canonical Shh targets in human SHH-MB. Genes from multiple human (pink) and mouse (grey) Shh-MB datasets were ranked according to the degree to which they are overexpressed in Shh-MB compared to other MB groups (human) or healthy cerebellar tissue (mouse). Each point represents the quantile rank in one dataset. Boxplots show the median and interquartile range for each gene/species pair. Canonical Shh target genes *Gli1/GLI1*, *Ptch1/PTCH1, Ccnd1/CCND1*, and *MycN/MYCN* all rank highly in both mouse and human datasets, whereas Gpr153 ranks highly in all mouse datasets, but ranks in the lower half of all overexpressed genes in human SHH-MB datasets. (**B**) Gene expression patterns in Shh-MB are very similar among mouse datasets and among human datasets but show a limited correlation between human and mouse datasets. Expression data derived from microarray analysis were downloaded from GEO and reanalyzed. Correlogram [34] of median gene upregulation values in Shh-MB (see methods) between every two datasets is shown. The lower panel shows concentration ellipses and loess smoothed curves; the upper panel shows Pearson correlation coefficients in boxes shaded according to correlation coefficient values (blue-positive, red-negative).

**Figure 4 cells-08-00216-f004:**
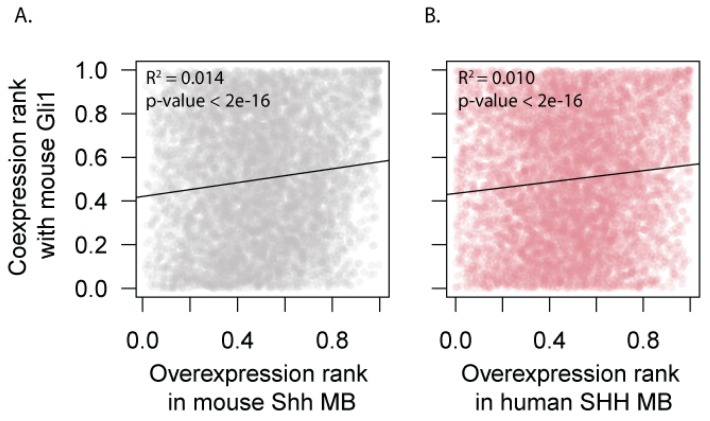
Mouse Hedgehog (HH) target genes are upregulated in both human and mouse Shh-MB. Genes were ranked according to their likelihood of being Shh targets in the mouse based on their coexpression patterns with the canonical Shh target Gli1 (y-axis) and ranked according to their overexpression in mouse Shh-MB (x-axis in **A**) or in human SHH-MB (x-axis in **B**). The rank R2 value is significantly different from 0 in both cases.

**Figure 5 cells-08-00216-f005:**
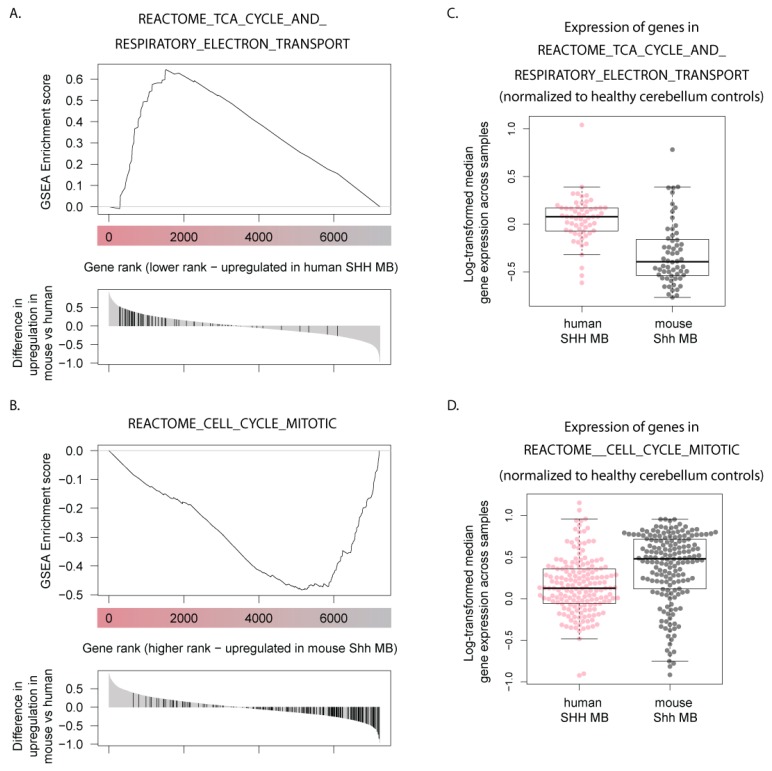
(**A**,**B**) Genes specifically upregulated in human SHH-MB (**A**) and mouse Shh-MB (**B**) are enriched for genes associated with mitochondrial function and proliferation, respectively. Gene set enrichment analysis (GSEA preranked) was performed on a list of genes ordered according to differences between their overexpression values in human vs. mouse Shh-MB. Example GSEA plots from among the most highly significant categories are presented. See also Table 2 and Table 3. (**C**,**D**) Median log-transformed expression of genes from the two GSEA gene sets shown in A. and B. in human and mouse SHH-MB compared to their respective healthy cerebellum controls. Cerebellum expression values equal 0 in both cases. Each data point represents a different gene from the gene set. See also Appendix A.

**Figure 6 cells-08-00216-f006:**
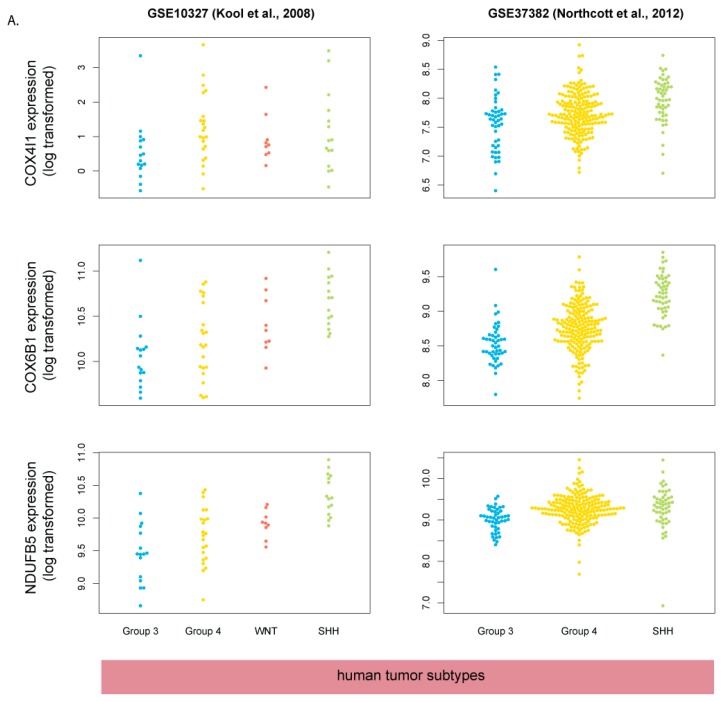
Mitochondrial respiratory chain genes are upregulated in human SHH-MB (**A**) but not in mouse Shh-MB (**B**). Expression data derived from microarray analysis were downloaded from GEO and processed as in Figure 2.

**Figure 7 cells-08-00216-f007:**
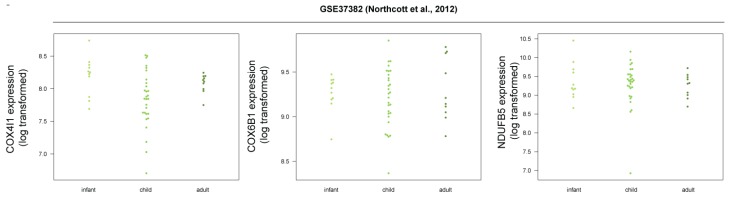
Mitochondrial respiratory chain genes do not show different gene expression patterns in human SHH-MB depending on patient age group. Expression data derived from microarray analysis were downloaded from GEO and processed as in Figure 2.

**Figure 8 cells-08-00216-f008:**
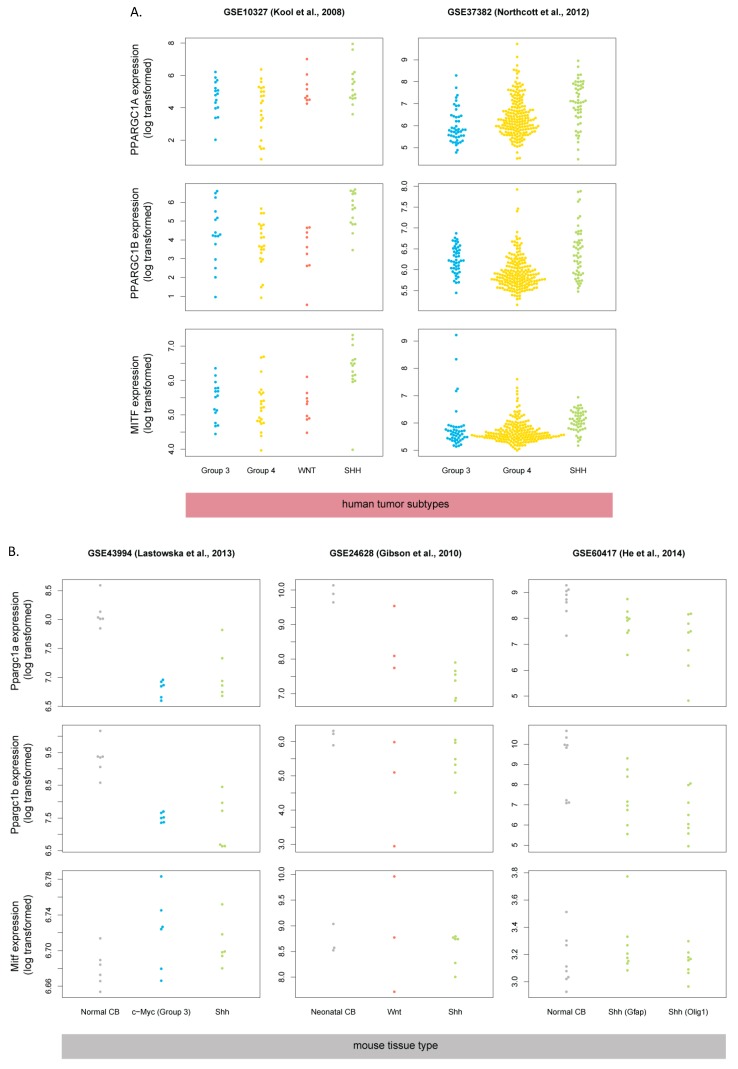
Oxidative phosphorylation master regulator genes are upregulated in human SHH-MB (**A**) but not in mouse Shh-MB (**B**). Expression data derived from microarray analysis were downloaded from GEO and processed as in Figure 2.

**Figure 9 cells-08-00216-f009:**
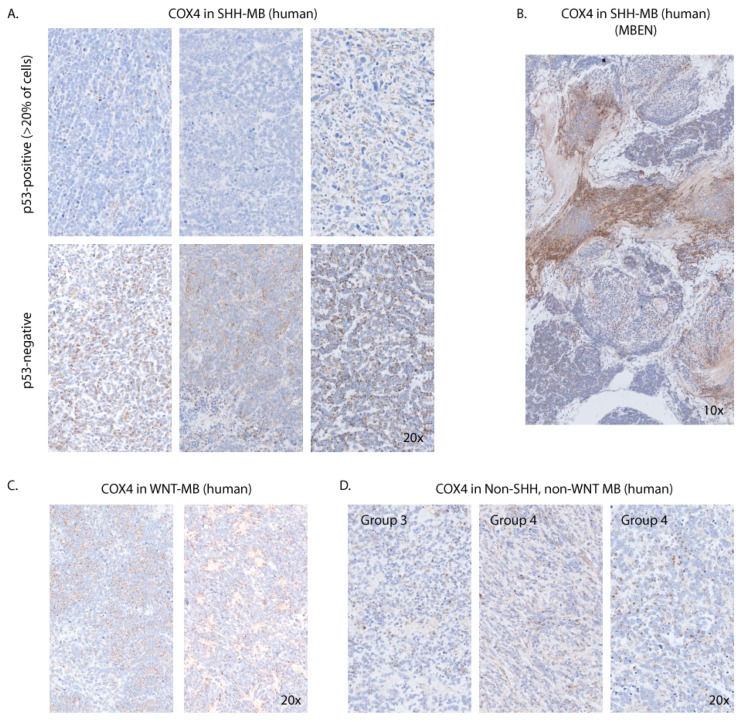
COX4 is highly expressed in a subset of human SHH-MB. Sections of FFPE tumor samples were stained with the anti-COX4 antibody (brown) and counterstained with hematoxylin (blue). Images were scanned at an original magnification of 40×. Digital magnification is 20× for all images except MBEN-type SHH-MB (**B**). (**A**) COX4 in primary SHH-MB tumors according to p53 status. (**B**) MBEN-type SHH-MB showing strong COX4 immunoreactivity in the internodular region imaged at 10× physical magnification. (**C**) WNT-MB tumors imaged at 20× physical magnification. (**D**) Non-SHH-, non-WNT-type tumors imaged at 20× magnification. The molecular subgroup is indicated for each sample.

**Figure 10 cells-08-00216-f010:**
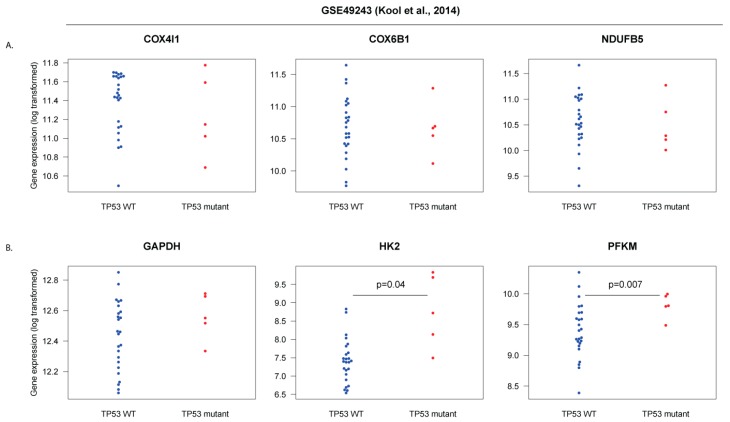
Expression of markers of mitochondrial oxidative phosphorylation (**A**) and glycolysis (**B**) in SHH-medulloblastoma (MB) tumors genotyped for mutations in the *TP53* gene. Microarray data from the GEO dataset GSE49243 was cross-referenced with supplemental data from [7] to obtain expression data in *TP53* WT and *TP53* mutant subgroups. Log2 expression is shown for each subgroup for the indicated genes.

**Table 1 cells-08-00216-t001:** Microarray datasets used in transcriptomic analyses.

GEO Accession Number	Species	Microarray Platform	Ref.	Dataset Description
GSE60417	mouse	Affymetrix Mouse Gene 1.1 ST Array	[12]	Shh-MB model: Gαs cKO in *Olig1*- or *GFAP*-expressing cells
GSE9299	mouse	Affymetrix Mouse Expression 430A/B Array Set	[29]	Shh-MB model: postnatal SmoM2 overexpression
GSE43994	mouse	Illumina MouseRef-8 v2.0 expression beadchip	[30]	Shh-MB model: *Ptch1*^+/−^
GSE24628	mouse	Affymetrix Mouse Genome 430 2.0 Array	[28]	Shh-MB and Wnt-MB model: *Ptch1*^+/−^; *Tp53*^−/−^ and *Blbp-Cre*^+/−^; *Ctnnb1*^+/*lox(ex3)*^; *Tp53^flx/flx^*
GSE37418	human	Affymetrix Human Genome U133 Plus 2.0 Array	[31]	human tumors of different subtypes (Wnt, Shh, Group 3, Group 4)
GSE41842	human	Affymetrix Human Gene 1.0 ST Array	[32]	human tumors of different subtypes (Wnt, Shh, Group 3, Group 4)
GSE10327	human	Affymetrix Human Genome U133 Plus 2.0 Array	[27]	human tumors of different subtypes (Wnt, Shh, Group 3(C), Group 4(D), Group E)
GSE37382	human	Affymetrix Human Gene 1.1 ST Array	[26]	human tumors of different subtypes (Wnt, Shh, Group 3)
GSE28245	human	Agilent-014850 Whole Human Genome Microarray 4x44K G4112F	[33]	human tumors of different subtypes (Wnt, Shh, Group 3, Group 4)
GSE49243	human	Affymetrix Human Genome U133 Plus 2.0 Array	[7]	human SHH-MB tumors with matched mutation data

**Table 2 cells-08-00216-t002:** Top Gene Set Enrichment Analysis (GSEA) categories for genes upregulated in human but not mouse SHH-MB (FWER—family-wise error rate).

Gene Set Name	FWER *p*-val
REACTOME_RESPIRATORY_ELECTRON_TRANSPORT_ATP_SYNTHESIS_BY_CHEMIOSMOTIC_COUPLING_AND_HEAT_PRODUCTION_BY_UNCOUPLING_PROTEINS_	0
REACTOME_TCA_CYCLE_AND_RESPIRATORY_ELECTRON_TRANSPORT	0
REACTOME_RESPIRATORY_ELECTRON_TRANSPORT	0
KEGG_OXIDATIVE_PHOSPHORYLATION	0
POMEROY_MEDULLOBLASTOMA_DESMOPLASIC_VS_CLASSIC_DN	0
MOOTHA_VOXPHOS	0
LEE_TARGETS_OF_PTCH1_AND_SUFU_DN	0
ANASTASSIOU_MULTICANCER_INVASIVENESS_SIGNATURE	0
GO_RESPIRATORY_CHAIN	0
GO_REGULATION_OF_POSTSYNAPTIC_MEMBRANE_POTENTIAL	0
LEIN_NEURON_MARKERS	0.001
VECCHI_GASTRIC_CANCER_EARLY_DN	0.001
DELYS_THYROID_CANCER_DN	0.005
GO_CELLULAR_RESPIRATION	0.007
GO_MITOCHONDRIAL_PROTEIN_COMPLEX	0.008

**Table 3 cells-08-00216-t003:** Top Gene Set Enrichment Analysis (GSEA) categories for genes upregulated in mouse but not human SHH-MB (FWER—family-wise error rate).

Gene Set Name	FWER *p*-val
HALLMARK_E2F_TARGETS	0
REACTOME_CELL_CYCLE	0
REACTOME_CELL_CYCLE_MITOTIC	0
MARSON_BOUND_BY_E2F4_UNSTIMULATED	0
REACTOME_DNA_REPLICATION	0
POMEROY_MEDULLOBLASTOMA_DESMOPLASIC_VS_CLASSIC_UP	0
SOTIRIOU_BREAST_CANCER_GRADE_1_VS_3_UP	0
ZHANG_TLX_TARGETS_60HR_DN	0
CHANG_CYCLING_GENES	0
REACTOME_S_PHASE	0
REACTOME_SYNTHESIS_OF_DNA	0
NIKOLSKY_BREAST_CANCER_17Q21_Q25_AMPLICON	0
GO_POSTREPLICATION_REPAIR	0
KONG_E2F3_TARGETS	0.005
PUJANA_XPRSS_INT_NETWORK	0.005
FISCHER_DREAM_TARGETS	0.005
ROSTY_CERVICAL_CANCER_PROLIFERATION_CLUSTER	0.008

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
