# Peer review of "Differential Expression of Mitochondrial Biogenesis Markers in Mouse and Human SHH-Subtype Medulloblastoma"

_cells, 2019, doi:10.3390/cells8030216_

Round 1

Reviewer 1 Report

This very interesting paper builds on previous work by the authors identifying a novel Shh target gene using mouse models of Shh MBB but shown not to upregulated in human tissues. Here the authors ask a comprehensive series of questions to investigate similarities and differences between human tumour and mouse models of Shh MBB via generation of novel datasets and analysis of online datasets. They conclude that differences are due to differences in pathophysiology and not due to differences in the signalling within the Shh pathway itself in the two species. They go on to identify differences in glycolytic metabolism (higher in mouse models than human tissues), including investigating protein expression and localisation of differentially-expressed genes.  

Identification of such differences between mouse models and human patient samples is important in ensuring that appropriate models are used for target identification and validation, and for pre-clinical drug screening. Given the identified problem with the mouse syngeneic models currently used for such screening, it might be helpful if the authors commented on alternative models that could be used for this purpose.

Author Response

Thank you for the critical reading of our manuscript and for helpful suggestions. We respond to the comments below.

This very interesting paper builds on previous work by the authors identifying a novel Shh target gene using mouse models of Shh MBB but shown not to upregulated in human tissues. Here the authors ask a comprehensive series of questions to investigate similarities and differences between human tumour and mouse models of Shh MBB via generation of novel datasets and analysis of online datasets. They conclude that differences are due to differences in pathophysiology and not due to differences in the signalling within the Shh pathway itself in the two species. They go on to identify differences in glycolytic metabolism (higher in mouse models than human tissues), including investigating protein expression and localisation of differentially-expressed genes. 

Identification of such differences between mouse models and human patient samples is important in ensuring that appropriate models are used for target identification and validation, and for pre-clinical drug screening. Given the identified problem with the mouse syngeneic models currently used for such screening, it might be helpful if the authors commented on alternative models that could be used for this purpose.

We have included a discussion of alternative preclinical models of MB in the discussion section.

Reviewer 2 Report

The manuscript proposes that genes involved in mitochondrial biogenesis are up-regulated in patient-derived samples of SHH-driven medulloblastoma, and that this up-regulation represents a point of variance between the human disease and the transgenic mouse models that are often used to study it. Based on this proposed difference, the authors argue that mouse models of SHH-driven medulloblastoma are likely to have different energy metabolism than of SHH-driven medulloblastoma in humans. There are major flaws in the analysis that undermine these conclusions.

The results sections starts with a comparison of gene expression in cerebellar granule cell progenitors treated with either SHH or with SHH AND the PkA activator PACAP, which they authors have shown to inhibit SHH-driven proliferation. The authors state the hypothesis that “genes that were upregulated in Shh-treated cGCPs but were downregulated by the addition of PACAP could serve as markers of Shh activity in MB precursor cells and potential biomarkers/therapy targets for human SHH-MB”. A relatively minor concern is that the rationale for this hypothesis is not obvious and should be explained. Specifically, the authors need to make clear why they expect the comparison of SHH versus SHH+PACAP to be more informative than the comparison of SHH versus no SHH. Another relatively minor concern is that it is not clear whether this comparison is a new experiment, or is simply an analysis of data from their prior published paper, reference 14.

The comparison identifies a set of differentially expressed genes, of which the authors pick two, Gpr153 and Shisa2. As relatively minor issues, the full set of genes is not presented in this paper (although it may be in the prior reference – if so, the text should make clear) and the basis for picking these two genes from the list is not adequately described.

The key observation reported by the authors regarding Gpr153 and Shisa2 is that these genes are consistently up-regulated in mouse models of SHH-driven medulloblastoma, but do not seem to show differentially high gene expression in patient-derived SHH-subtype medulloblastoma compared to other subtypes of medulloblastoma. However, the use of other medulloblastoma subtypes as the basis for comparison is a major flaw, because it is possible that GPR153 and/or SHISA2 may be highly expressed in BOTH SHH-driven medulloblastoma and other medulloblastoma subtypes. If so, normalizing to other medulloblastoma subtypes will falsely seem to show that GPR153 and/or SHISA2 are not up-regulated. Put another way, the facts that GPR153 and SHISA2 are not expressed at relatively higher levels in SHH-subgroup tumors compared to Group 3 tumors do not exclude the possibility that these genes are expressed at high levels in SHH-subgroup tumors. If microarray datasets of mouse and human tumor analyses are to be compared, the same type of reference tissue must be used for comparison in all datasets. It is not acceptable to compare relative expression values for mouse tumors to normal mouse brain to relative expression values for human tumors compared to other human tumor types.  

An alternative way to determine if GPR153 and SHISA2 and proteins are expressed in human medulloblastoma would be immunohistochemistry. To accomplish this, the authors should est whether antibodies to GPR153 and SHISA2 label SHH-subgroup medulloblastoma sections and shuld include relevant human positive and negative control tissue. Without addressing these issues, the premise, stated directly as “Thus, increased expression of Gpr153 and Shisa2 is a characteristic feature of mouse Shh-MB tumors, but not human SHH-MBs” is not supported by the evidence presented.

The same issue of how to make relative comparisons in a systematic way using mouse and human datasets compromises the more general conclusion that SHH-driven medulloblastomas in mice do not closely resemble SHH-subgroup medulloblastomas in humans, as portrayed most directly in Figure 4B. The authors do not make fully clear exactly what denominators are used to normalize expression data in the mouse and human datasets, beyond the comment “Next, we normalized each row by subtracting the mean value for that row from all values within the row (normalized median gene expression values). This generated data that allowed us to determine whether median expression of a gene in a specific tumor/tissue type is higher (positive values) or lower (negative values) from other tumor/tissue types in the same dataset”. What is missing from the description is what was in each column of the dataset—specifically, were the mouse SHH tumors and the human SHH tumors compared to the same type of internal controls, or was there a systematic difference in the reference “tumor/tissue type” used in the mouse and human studies. If mouse studies tended to use normal adult cerebellum as the reference tissue, while human studies tended to use WNT, Group3 and Group4 tumors as the reference tissue, it would be expected that there would be internal cohesion when the mouse datasets and compared to each other, and when the human datasets are compared to each other, and a general lack of similarity when the mouse datasets are compared with the human datasets. In order to the mouse and human datasets to be comparable, they must be normalized against a common reference tissue. Otherwise, comparison between cannot be used to support the conclusions the authors draw.

This same concern undermines the GSEA data. It is predictable that if the reference tissue in the mouse studies is non-mitotic tissue, that GSEA would show up-regulation of gene related to cell cycle progression and RB function. In contrast, if the human data represent comparison of gene expression in SHH-subgroup tumors vs other medulloblastoma subgroups, which are all highly proliferative, it is predictable that cxell cycle-regulating genes will not be differentially represented. If the reference tissues are not highly similar, not inference across datasets can be made.

In the final figure, the authors argue that their COX4 staining shows the relative importance of mitochondrial biogenesis in human SHH subgroup medulloblastoma. However, they do not present immunohistochemistry data showing COX4 mouse SHH-driven medulloblastoma. Without comparison to the mouse tumors, these data cannot be used to argue that mitochondrial biogenesis differentiates human SHH subgroup medulloblastoma from the SHH-driven mouse models.

Another issue that is not considered is that the expression of mitochondrial genes does not preclude the possibility that tumors use aerobic glycolysis. Cells may use oxidative phosphorylation and aerobic glycolysis at the same time. Aerobic glycolysis is demonstrated by the production of lactate from glucose in the presence of oxygen. There is no requirement for mitochondrial inactivity or lack of expression of genes related to oxidative phosphorylation. As a result, even if the data did show that human SHH subgroup medulloblastomas express mitochondrial genes that were not up-regulated in mouse SHH-driven models (which is NOT demonstrated by the current data), these data would not be sufficient to show that these tumors use different mechanisms of energy generation. Some discussion of this nuance is needed.

Given the important role of COX4 in homeostasis, it is quite interesting that the authors have found heterogeneity of COX4 expression in human tumors and markedly reduced COX4 staining in SHH subgroup tumors with p53 immunoreactivity. These data are striking, even if they do not support the overall proposal of the paper.

The logical flaws in the analysis are critical. The paper should not be published unless the issue of using a common reference tissue across datasets can be resolved. There is a significant chance that with a common reference tissue used as the denominator across mouse and human datasets, the data will indicate very different conclusions.

Author Response

Thank you for the critical reading of our manuscript and for helpful suggestions. We respond to the comments below.

The manuscript proposes that genes involved in mitochondrial biogenesis are up-regulated in patient-derived samples of SHH-driven medulloblastoma, and that this up-regulation represents a point of variance between the human disease and the transgenic mouse models that are often used to study it. Based on this proposed difference, the authors argue that mouse models of SHH-driven medulloblastoma are likely to have different energy metabolism than of SHH-driven medulloblastoma in humans. There are major flaws in the analysis that undermine these conclusions.

 The results sections starts with a comparison of gene expression in cerebellar granule cell progenitors treated with either SHH or with SHH AND the PkA activator PACAP, which they authors have shown to inhibit SHH-driven proliferation. The authors state the hypothesis that “genes that were upregulated in Shh-treated cGCPs but were downregulated by the addition of PACAP could serve as markers of Shh activity in MB precursor cells and potential biomarkers/therapy targets for human SHH-MB”. A relatively minor concern is that the rationale for this hypothesis is not obvious and should be explained.  Specifically, the authors need to make clear why they expect the comparison of SHH versus SHH+PACAP to be more informative than the comparison of SHH versus no SHH.

PACAP acts through its cognate GPCR receptors to induce Gαs and therefore induce protein kinase A by upregulation of adenylate cyclase activity. PACAP is a physiological inhibitor of Shh signaling and MB formation, and acts at the level of Gli proteins. Therefore, genes that respond to both Shh and PACAP are more likely to be of physiological and pathological significance than genes that respond only to Shh.

Another relatively minor concern is that it is not clear whether this comparison is a new experiment, or is simply an analysis of data from their prior published paper, reference 14.

This is a reanalysis of the data generated as part of a prior published paper (reference 14). We have changed the relevant sentence to reflect that fact.

The comparison identifies a set of differentially expressed genes, of which the authors pick two, Gpr153 and Shisa2. As relatively minor issues, the full set of genes is not presented in this paper (although it may be in the prior reference – if so, the text should make clear) and the basis for picking these two genes from the list is not adequately described.

We addressed these two issues in the text.

The key observation reported by the authors regarding Gpr153 and Shisa2 is that these genes are consistently up-regulated in mouse models of SHH-driven medulloblastoma, but do not seem to show differentially high gene expression in patient-derived SHH-subtype medulloblastoma compared to other subtypes of medulloblastoma. However, the use of other medulloblastoma subtypes as the basis for comparison is a major flaw, because it is possible that GPR153 and/or SHISA2 may be highly expressed in BOTH SHH-driven medulloblastoma and other medulloblastoma subtypes. If so, normalizing to other medulloblastoma subtypes will falsely seem to show that GPR153 and/or SHISA2 are not up-regulated. Put another way, the facts that GPR153 and SHISA2 are not expressed at relatively higher levels in SHH-subgroup tumors compared to Group 3 tumors do not exclude the possibility that these genes are expressed at high levels in SHH-subgroup tumors. If microarray datasets of mouse and human tumor analyses are to be compared, the same type of reference tissue must be used for comparison in all datasets. It is not acceptable to compare relative expression values for mouse tumors to normal mouse brain to relative expression values for human tumors compared to other human tumor types. 

We address the issue of a common reference sample below

An alternative way to determine if GPR153 and SHISA2 and proteins are expressed in human medulloblastoma would be immunohistochemistry. To accomplish this, the authors should est whether antibodies to GPR153 and SHISA2 label SHH-subgroup medulloblastoma sections and shuld include relevant human positive and negative control tissue. Without addressing these issues, the premise, stated directly as “Thus, increased expression of Gpr153 and Shisa2 is a characteristic feature of mouse Shh-MB tumors, but not human SHH-MBs” is not supported by the evidence presented.

We do not have validated antibodies against GPR153 and SHISA2. We did not intend to make statements about the expression of GPR153 and SHISA2 on the protein level, but rather on the mRNA level. This may have been confusing because we did not use italics to signify gene expression. We have now corrected this. For GPR153 we have now confirmed reduced expression not only compared to all other MB subgroups, but also compared to healthy CB (see below for detailed explanation). For SHISA2 such analysis was not possible, because SHISA2 expression is not measured in the combined healthy CB/MB dataset that we used (image in the attached Word file).

The same issue of how to make relative comparisons in a systematic way using mouse and human datasets compromises the more general conclusion that SHH-driven medulloblastomas in mice do not closely resemble SHH-subgroup medulloblastomas in humans, as portrayed most directly in Figure 4B. The authors do not make fully clear exactly what denominators are used to normalize expression data in the mouse and human datasets, beyond the comment “Next, we normalized each row by subtracting the mean value for that row from all values within the row (normalized median gene expression values). This generated data that allowed us to determine whether median expression of a gene in a specific tumor/tissue type is higher (positive values) or lower (negative values) from other tumor/tissue types in the same dataset”. What is missing from the description is what was in each column of the dataset—specifically, were the mouse SHH tumors and the human SHH tumors compared to the same type of internal controls, or was there a systematic difference in the reference “tumor/tissue type” used in the mouse and human studies. If mouse studies tended to use normal adult cerebellum as the reference tissue, while human studies tended to use WNT, Group3 and Group4 tumors as the reference tissue, it would be expected that there would be internal cohesion when the mouse datasets and compared to each other, and when the human datasets are compared to each other, and a general lack of similarity when the mouse datasets are compared with the human datasets. In order to the mouse and human datasets to be comparable, they must be normalized against a common reference tissue. Otherwise, comparison between cannot be used to support the conclusions the authors draw.

This was indeed a very important limitation of our original manuscript, and we have struggled with a way to address this. The reviewer is correct in that at the time we were unable to match human and mouse SHH-MBs with identical controls. The reason was that the vast majority of datasets available for human samples contain either samples of healthy cerebella, or samples of MB tumors, but not both. The opposite is true for mouse MB datasets – these usually include healthy wild-type cerebellum controls, but are usually limited to one MB subtype. We reasoned that using non-SHH MB samples as controls for SHH-MB tumors was justified, because the different tumor types were likely to have very different expression profiles that would cancel each other out, and genes that were consistently upregulated in SHH-MB would rank highly in our calculations independent of whether the controls were healthy cerebellar tissues or MB samples of different type. However, at the time we could not verify our assumptions.

Fortunately, a paper published after we had submitted our original manuscript allowed us to perform just the kind of analysis that this reviewer requests. The paper (Weishaupt et al., 2019, ref. [20])describes a novel way to combine microarray datasets containing healthy cerebellar controls and datasets containing MB tumors only. We used the dataset that was published alongside this paper to validate our assumption that using different MB subtypes as controls for SHH-MB was approximately equivalent to using healthy cerebella as controls. Specifically, we compared differences in ranks between human and mouse SHH-MB under two different analysis conditions. In one analysis, we calculated differences in ranks as before, using non-SHH MB as controls for human SHH-MB and using healthy CB or healthy CB + non-SHH MB as controls for mouse SHH-MB (depending on the dataset). In the other analysis, we ranked genes in the human based on the differences between SHH-MB and healthy CB controls in the combined Weishaupt et al. dataset, and we ranked genes in the mouse based on differences between SHH-MB and healthy CB controls in individual datasets. Importantly, it turned out that the differences in ranks were highly correlated whether we used non-SHH MB controls or healthy CB controls for human data (Supplemental Figure S2). Because our subsequent analyses are based on differences in ranks between human and mouse SHH-MB, these also remain unchanged whether we use non-SHH MB or CB as controls.

This same concern undermines the GSEA data. It is predictable that if the reference tissue in the mouse studies is non-mitotic tissue, that GSEA would show up-regulation of gene related to cell cycle progression and RB function. In contrast, if the human data represent comparison of gene expression in SHH-subgroup tumors vs other medulloblastoma subgroups, which are all highly proliferative, it is predictable that cxell cycle-regulating genes will not be differentially represented. If the reference tissues are not highly similar, not inference across datasets can be made.

Since differences in gene ranks between human and mouse SHH-MB are largely preserved independent of controls used (Figure S2), we reasoned that GSEA results would also remain the same. Indeed, GSEA performed on ranks calculated using CB as controls are largely the same as GSEA calculated previously. Specifically, the main conclusion of the paper still stands: oxidative phosphorylation genes are on average slightly overexpressed in human SHH-MB compared to cerebellar controls, whereas they are strongly downregulated in mouse SHH-MB (see additional panel C in Fig. 5 and Fig. S3A). Similarly, expression of cell cycle genes is only slightly upregulated in human SHH-MB versus human CB, but is very strongly upregulated in mouse SHH-MB versus mouse CB (new panel D in Fig. 5, Fig. S3B).

In the final figure, the authors argue that their COX4 staining shows the relative importance of mitochondrial biogenesis in human SHH subgroup medulloblastoma. However, they do not present immunohistochemistry data showing COX4 mouse SHH-driven medulloblastoma. Without comparison to the mouse tumors, these data cannot be used to argue that mitochondrial biogenesis differentiates human SHH subgroup medulloblastoma from the SHH-driven mouse models.

We agree that the analysis of mouse Shh-MB tumors would be interesting. However, we currently do not have Shh-MB mice available to us, and obtaining and analyzing such samples would be impossible in the short time span required for resubmission of the manuscript. On the other hand, Di Magno et al., 2014 convincingly show using multiple assays that mouse Shh-MB are strongly glycolytic, and we also show that not just one, but the vast majority of mitochondrial oxidative phosphorylation genes are downregulated in Shh-MB at the mRNA level compared to healthy cerebellum controls (see the newly added panel C in Fig. 5). Given these results, we believe that the main conclusions of the paper are well justified even in the absence of mouse Shh-MB staining.

Another issue that is not considered is that the expression of mitochondrial genes does not preclude the possibility that tumors use aerobic glycolysis. Cells may use oxidative phosphorylation and aerobic glycolysis at the same time. Aerobic glycolysis is demonstrated by the production of lactate from glucose in the presence of oxygen. There is no requirement for mitochondrial inactivity or lack of expression of genes related to oxidative phosphorylation. As a result, even if the data did show that human SHH subgroup medulloblastomas express mitochondrial genes that were not up-regulated in mouse SHH-driven models (which is NOT demonstrated by the current data), these data would not be sufficient to show that these tumors use different mechanisms of energy generation. Some discussion of this nuance is needed.

We have added a relevant statement in the discussion.

Given the important role of COX4 in homeostasis, it is quite interesting that the authors have found heterogeneity of COX4 expression in human tumors and markedly reduced COX4 staining in SHH subgroup tumors with p53 immunoreactivity. These data are striking, even if they do not support the overall proposal of the paper.

The logical flaws in the analysis are critical. The paper should not be published unless the issue of using a common reference tissue across datasets can be resolved. There is a significant chance that with a common reference tissue used as the denominator across mouse and human datasets, the data will indicate very different conclusions.

Thank you for emphasizing these issues. We believe that we have sufficiently addressed this comment using the Weishaupt et al. dataset., which makes us even more confident in the conclusions of the paper.

Reviewer 3 Report

The authors of the manuscript titled “Differential expression of mitochondrial biogenesis markers in mouse and human SHH-subtype medulloblastoma” investigated the differences between the sonic hedgehog medulloblastoma mouse models and the human disease and found profound differences. In particular, it was found that p53-mutant human SHH medulloblastoma increased the expression of OXPHOS genes and that this was not the case in mouse. Together, the authors conclude that this finding may impact therapeutic intervention.

I find the manuscript well written, the data convincing, original, and interesting. I do not have any concerns however; the authors should include analysis of medulloblastoma cell culture models and report how these recapitulate or not, their finding in disease.

Author Response

The authors of the manuscript titled “Differential expression of mitochondrial biogenesis markers in mouse and human SHH-subtype medulloblastoma” investigated the differences between the sonic hedgehog medulloblastoma mouse models and the human disease and found profound differences. In particular, it was found that p53-mutant human SHH medulloblastoma increased the expression of OXPHOS genes and that this was not the case in mouse. Together, the authors conclude that this finding may impact therapeutic intervention.

I find the manuscript well written, the data convincing, original, and interesting. I do not have any concerns however; the authors should include analysis of medulloblastoma cell culture models and report how these recapitulate or not, their finding in disease.

Analysis of cell culture models would be complicated in this context. The cell lines currently in use in preclinical studies have serious drawbacks, the foremost among them being a serious genetic and phenotypic divergence from their tumors of origin, and adaptation to non-physiological cell culture conditions. We are not aware of any comparative studies of oxidative phosphorylation in human SHH MB lines, but even if such data exists, we should be very cautious in interpreting these results. In response to this reviewer’s and the first reviewer’s comments, we provide an overview of alternative models of SHH MB in the Discussion section.

Round 2

Reviewer 2 Report

The authors have ably addressed most of the issues cited in my review. I remain concerned that the down-regulation of COX4 in mouse medulloblastomas shoudl be demonstrated by immunohistochemistry. I am not convinced that doing this experiment would be a hardship. Many labs around the world regularly generate mouse medulloblastoma tissue and if asked would be likely to share paraffin sections or frozen sections for he authors to analzye COX4 staining. Additionally, COX4 antibodies are readily avaialble. The risk of publishing the data without this simple confirmatory step is that someone else may do the experiment and find out that COX4 is not down-regulated, and this aspect of the paper will turn out to have been wrong. I have no knowledge that it is wrong, but I am quite sure it would not be hard to test. This test would strengthen the quality of the work.

Author Response

Thank you again for providing constructive feedback on our manuscript. We agree that in principle it would be worthwhile to include the experiment you propose in the manuscript. In fact, we tried to find tissue that would be suitable to do the work in our freezers, but unfortunately it turned out that it had been discarded some time ago and was no longer available. We do not agree, however, that the proposed experiments would not be a hardship. They would be both costly and time-consuming, and we strongly believe that they would not greatly change the overall conclusions of the paper. First, performing these experiments would necessitate collaborating with a new research group, which would require significant logistical investment on our part. Second, there is no guarantee that the antibody that we used on human FFPE samples would work equally well on cryopreserved tissue embedded in OCT, which is more typical for mouse samples. Third, the antibody that we used was only tested in human, so using it on mouse samples might not work, especially since the antibody was raised in mouse and there is a high risk of non-specific cross-reactivity with mouse IgG in tumor blood vessels. Fourth, the time frame for resubmission on a minor revision is 5 days, which is not realistic for experimental work, unless all the materials are already at hand and require no additional optimization. Fifth, we would run into the same issue that this reviewer pointed out to us earlier – the controls would be different in mouse and in human, so the results would not be directly comparable. Finally, as we had emphasized in our previous response, the fact that mouse Shh-MB tumors are primarily glycolytic has already been shown using multiple assays by Di Magno et al. This in itself does not guarantee that Cox4 protein is lower in the mouse tumors than in surrounding tissue, but such confirmatory experiment would not in our view be worth the cost and effort.